# Dabrafenib, idelalisib and nintedanib act as significant allosteric modulator for dengue NS3 protease

R. V. Sriram Uday[1☯], Rajdip Misra[2☯], Annaram Harika[1], Sandip Dolui[2], Achintya Saha[3], Uttam Pal[2¤]*, V. Ravichandiran[1]*, Nakul C. Maiti[2]*

**1** National Institute of Pharmaceutical Education and Research, Kolkata, Chunilal Bhawan, Kolkata, West Bengal, India, **2** Structural Biology and Bioinformatics Division, Indian Institute of Chemical Biology, Council of Scientific and Industrial Research, Kolkata, West Bengal, India, **3** Department of Chemical Technology, University of Calcutta, Kolkata, West Bengal, India

☯ These authors contributed equally to this work.
¤ Current address: S.N. Bose National Centre for Basic Sciences, Salt Lake City, Kolkata, West Bengal, India
* ncmaiti@iicb.res.in (NCM); director@niperkolkata.edu.in (VR); uttam2707@gmail.com (UP)

**Data Availability Statement:** All relevant data are within the manuscript and its Supporting information files.

## Abstract

Dengue virus (DENV) encodes a unique protease (NS3/NS2B) essential for its maturation and infectivity and, it has become a key target for anti-viral drug design to treat dengue and other flavivirus related infections. Present investigation established that some of the drug molecules currently used mainly in cancer treatment are susceptible to bind non-active site (allosteric site/ cavity) of the NS3 protease enzyme of dengue virus. Computational screening and molecular docking analysis found that dabrafenib, idelalisib and nintedanib can bind at the allosteric site of the enzyme. The binding of the molecules to the allosteric site found to be stabilized via pi-cation and hydrophobic interactions, hydrogen-bond formation and π-stacking interaction with the molecules. Several interacting residues of the enzyme were common in all the five serotypes. However, the interaction/stabilizing forces were not uniformly distributed; the π-stacking was dominated with DENV3 proteases, whereas, a charged/ionic interaction was the major force behind interaction with DENV2 type proteases. In the allosteric cavity of protease from DENV1, the residues Lys73, Lys74, Thr118, Glu120, Val123, Asn152 and Ala164 were involved in active interaction with the three molecules (dabrafenib, idelalisib and nintedanib). Molecular dynamics (MD) analysis further revealed that the molecules on binding to NS3 protease caused significant changes in structural fluctuation and gained enhanced stability. Most importantly, the binding of the molecules effectively perturbed the protein conformation. These changes in the protein conformation and dynamics could generate allosteric modulation and thus may attenuate/alter the NS3 protease functionality and mobility at the active site. Experimental studies may strengthen the notion whether the binding reduce/enhance the catalytic activity of the enzyme, however, it is beyond the scope of this study.

**Funding:** The author(s) received no specific funding for this work.

**Competing interests:** The authors have declared that no competing interests exist.

**Abbreviations:** D2R, Dopamine D2 receptor; DENV Type 1–4, Dengue (type1, type 2, type 3 and type 4); DENV, Dengue virus; DENV-1, DENV-2, DENV-3, DENV-4, four different serotypes of the dengue virus:, Serotype, a serotype refers to a group of viruses classified together based on their antigens or surface proteins on the surface of the virus; DENV4T, Dengue Thailand variant; DHF, Dengue hemorrhagic fever; Ds, double-stranded; DSS, Dengue shock syndrome; HCV, Hepatitis C virus; HIV, Human immunodeficiency virus; kb, kilo bases; kd, Equilibrium dissociation constant; MD, Molecular dynamics; NS, nonstructural; NS2B, Non-structural protein 2B; NS2B/NS3, Dengue NS2B protein linked with NS3 as co-factor; NS3, Non-structural protein 3; NS3pro, Dengue NS3 protease domain; NS4, Non-structural protein 4; NS5, Non-structural protein 5; NTPase, Nucleoside triphosphatase; PCZ, Prochlorperazine; PDB, Protein data bank; $R_g$, Radius of gyration; RMSD, Root mean square deviation; RMSF, Root mean square fluctuation; RTPase, RNA triphosphatase; VS, Virtual screening.

## Introduction

Dengue virus (DENV) is a mosquito-borne human pathogen that causes dengue fever, dengue shock syndrome (DSS) and dengue hemorrhagic disease (DHD). The virus is of ~50-nm diameter, enveloped with a lipid membrane [1, 2]. The virus comes under the family of flaviviridae, genus of flavivirus and species Dengue virus [3]. There are four different serotypes of Dengue virus, namely DENV1, DEVN2, DENV3, and DENV4. Another new variant of DENV4, DENV4T was recently reported in 2013 [4]. The major difference for humans to different dengue types lies in subtle differences in the surface proteins of the different dengue serotypes. Infection induces long-life protection against the infecting serotype; however, it gives only a short time protective immunity against the other types. The primary infection causes minor disease, but secondary infections have been reported to cause severe diseases such as dengue hemorrhagic fever (DHF) or dengue shock syndrome (DSS) in both children and adults [5]. The recent survey estimated that about 390 million-dengue infections occur every year across the world [6].https://www.zotero.org/google-docs/?0uROhO However, a successful drug is yet to come out for the proper treatment of these diseases.

A significant effort is undergoing for the development of antiviral drugs for the cure of the disease. Two general strategies are often pursued for any antiviral therapy. The first is to inhibit viral targets; such an approach has generated most of the direct acting antivirals currently in clinical use. The second is to inhibit host targets. Two types of host targets could be pursued: (i) host factors that are essential for viral replication and (ii) host targets that participate in the development of disease symptoms [7]. Unfortunately, the molecular details of host pathways that lead to dengue fever are not well defined; therefore, the feasibility of such an approach remains low at this time and the major focus is on viral proteins and RNA molecules. The dengue viral RNA is an 11 kb long single stranded positively sense RNA which after translation synthesizes a single polyprotein chain. The polyprotein chain consists of three structural (capsid, membrane, envelope) and seven non-structural (NS1, NS2A, NS2B, NS3, NS4A, NS4B, NS5) proteins. In the polyprotein precursor molecule these proteins are arranged in a specific manner viz. $NH_2$– C-PrM-E-NS1-NS2A-NS2B-NS3-NS4A-NS4B-NS5-COOH [7, 8]. The structural proteins are associated with the construction of viral particle whereas the non-structural proteins mainly help in the viral RNA replication, assembly of virions as well as in the post translational processing.

Based on the dengue life cycle, one of the current drug targets in dengue drug discovery involves two main non-structural proteins NS3 and NS5 (Scheme 1) [9]. NS3 functions as protease and helicase, whereas NS5 is the RNA dependent RNA polymerase, in charge of viral RNA replication [10, 11]. Nonstructural protein 3(NS3) protease, coupled with cofactor NS2, is very essential for the replication process of the virus and it performs the task of post translational cleavage of the viral poly protein. Thus NS3 becomes an attractive target for anti-viral drug design. Several attempts has been made to develop small molecules as protease inhibitors targeting the NS3 active site [12–14]. However, the success rate was very limited and most of the inhibitors are peptide based. The active site of dengue NS3 protease is flat and it makes specific binding of a small molecule insignificant. Moreover, it binds preferentially substrates with basic subunits. Therefore, the inhibitors that bind another region of the protease and not at the NS3 active site appeared to be a better approach to design inhibitors.

**Scheme 1. Graphical representation of different structural and non-structural proteins of Dengue virus.** Top panel shows the scheme of the dengue viral polyprotein. C-capsid, prM-Membrane precursors, E-envelope, NS2B-Co-factor of NS3, NS3-Serine Protease and Helicase, NS5-RNA dependant RNA polymerase, Methyl Transferase. Lower panel shows the NS3 Protein with proteases and helicase parts. Active site of the protease is marked.

| Structural Proteins | | | Non-Structural Proteins | | | | | | |
|---|---|---|---|---|---|---|---|---|---|
| C | prM | E | NS1 | NS2A | NS2B | NS3 | NS4A | NS4B | NS5 |

NS2B/NS3 Protein

Protease and the catalytic triad

Earlier investigations suggested that the dengue virus protease family is susceptible to allosteric inhibition [15–18] and some of inhibitory antibodies utilize allosteric mechanisms in the inhibition processes of proteases. Wu et al. suggested diaryl-(thio)-ethers as candidates for a novel class of protease inhibitors [19]. They also proposed that these molecules participate in selective and non-competitive inhibition of the DENV2 and DENV3 cells by benzothiazole derivatives exhibiting IC50 values in the low micro molar range. Othman et al. found that group of flavanones and their chalcones, isolated from *Boesenbergia rotunda L*, can have varying degrees of noncompetitive inhibitory activities toward DENV2 protease [20]. Results obtained from automated docking studies are in agreement with experimental data in which the ligands were shown to bind to sites other than the active site of the protease i.e., allosteric sites. They have calculated $K_i$ values, which are very small, indicating that the ligands bind quite well to the allosteric binding site. Teixeira et al. reviewed on various plants, which are reported to be having anti DENV property [21]. Importance of plant flavonoids on inhibition of dengue virus was also reported by Hassandarvish et.al. The authors identified some flavonoid compounds, namely baicalein and baicalin from *Scutellaria baicalensi*, which showed promising inhibitory results against dengue viral replication. Allosteric inhibition of NS3 protease could be one of the several possible mechanisms for their antiviral activity as reported by the authors based on their in-silico docking studies [22]. From the concept of allosteric inhibition of enzymes, modern drug discovery also focused on the allosteric targets of the protein because of its more saturability and greater selectivity. The concept is, when the ligand binds to the allosteric site, it brings out some conformational changes in the protein and disables the

function of the protein. Several drugs have been developed for various targets, where the inhibition occured through targeting the allosteric sites e.g., benzodiazepines, barbiturates. Steroids target at allosteric sites of GABA receptor; aspirin, and sodium salicylates target Endothelin A receptors at allosteric sites etc. Allosteric site on the dengue protease was identified by blind docking followed by clustering as described by Pal et al. [23]. The importance of dynamics and fluctuation in the allosteric sites and its impacts on the functional efficacy of proteases in general discussed in previous reports [23, 24]. Upon small-molecule binding to the allosteric sites in the protein undergo a change in conformation or an alteration in the conformational equilibrium that affects the function of the enzyme.

A study [16], reported recently by Lim et.al, shows successful inhibition of dengue NS2B/NS3 protease by means of allosteric modulation. Similar type of approach was also reported earlier where authors have targeted dengue NS2B/NS3 by inhibiting its allosteric cavities [15, 17]. Thus allosteric inhibition of the viral protease stands a viable approach in anti-viral drug development. Therefore, by choosing DENV NS3 as target, we established by *in-silico* methods three molecules which could bind effectively to NS3, albeit at the allosteric site of the proteases, of all the five serotypes of the virus.

The protease domain in NS3 starts with N-terminal, which extends up to 180 amino acids (Scheme 1). The enzyme consists of 6 β-strands which form two β-barrels with the catalytic triad (His51, Asp75 and Ser135) [11]. Activity of the protease completely depends on the presence of its co-factor, NS2B that is conserved among almost all the flaviviruses. The NS3 protein includes residues 49–66 from the NS2B which were linked to the N-terminus of full-length NS3 by a Gly-Ser linker (Scheme 1). Protease domain is inactive without NS2B as it plays a critical role in proteolytic activation. The central region of NS2B (residues 67–80) interacts with the protease, flanking hydrophobic regions of NS2B are predicted to anchor the NS2B–NS3 complex in the ER membrane [25].

Drug repurposing nowadays is a cleaver tool which allow us to check the concealed potencies of pre-discovered drugs against various diseases. From past literatures we have seen some occasions of drug repurposing based approaches, where repurposed drug has been used to inhibit dengue infections. Yogy Simanjuntak et al. found that prochlorperazine (PCZ), a dopamine D2 receptor (D2R) antagonist approved to treat nausea, vomiting, and headache in humans has potent in vitro and in vivo antiviral activity against DENV infection [26]. They have targeted viral binding and viral entry through D2R and clathrin-associated mechanisms, which can be blocked by PCZ. However, in most of the scenario the target of the repurposed drugs is non-NS3 regions. In our present investigation we targeted NS3 protease and proceeded with a strategy of drug repurposing as a lot of protease and serine protease inhibitors are available in the market. We compiled a library of drugs for virtual screening, from which leads were identified and subjected for refinements. The molecules dabrafenib, idelalisib and nintedanib are identified as leads from drugs molecules library. The above lead molecules found to bind at the allosteric site of the target proteins of all the virus serotypes. Binding of the molecules to the allosteric site causes some conformational changes, reduces the motion as reflected in fluctuation and indicated possibility of terminating the protein function.

The techniques used in the investigation included molecular docking based virtual screening and molecular dynamics simulations. The leads we identified on screening libraries and later subjected to docking refinements and the best hits were subjected to molecular dynamic simulation studies. Molecular dynamics were performed by GROMACS using GROMOS96 53A6 force field with SPC water for 100 ns. Molecular dynamics analysis was made on NS3 (PDB: 3L6P) of serotype type 1 protein docked with three ligands individually. The structural and dynamic behavior of the protein when bound to ligand was monitored for duration of 100 ns. We analyzed structural deviations (RMSD; root mean square deviation), residue-wise

fluctuation (RMSF; root mean square fluctuation), flexibility ($R_g$; radius of gyration), protein-ligand interactions, and secondary structure analysis using DSSP tool. The results so obtained suggested that these molecules were potentially strong allosteric modulators of the NS3 activity and may act as candidates for the treatment of the disease state.

## Methods

### Multiple sequence alignment

Multiple sequence alignments is used in studies to characterize the proteins families, identify the shared regions of homology etc., in our studies the alignments are done in the Clustal Omega server which is a general purpose multiple sequence alignment program for DNA or proteins [27, 28]. The sequences of the five NS3 protease serotypes were obtained from Uni-Prot KnowledgeBase for the alignment. Percentage identity matrix was also obtained from the alignment.

### Selection of protein and small molecules for screening

Three-dimensional structural information about proteins is collected from Protein Data Bank (PDB). 24 crystal structures of different serotypes of dengue NS3 protein were found in PDB, from which best 13 crystal structures are selected for our studies that contained the protease domain. Small molecules are chosen from a library of small molecules that contains 3D structures of approved drugs of categories like protease inhibitors, ATP Kinase inhibitors etc. From this library by performing virtual screening using PyRX three drug molecules were selected. We selected drug molecules from the library of small molecules that contain 3D structures of approved drugs of various categories like protease inhibitors, ATP kinase inhibitors etc. (S1 Table). The 3D structures are collected from different databases like ChemSpider and Pub-Chem. Some structures are drawn in Avogadro software and geometry optimized performs prior to virtual screening. Subsequently, the molecules are subjected to energy minimization process using the steepest descent followed by conjugate gradient algorithms in UFF force field as implemented in Avogadro. The minimized small molecules, which are saved, converted into a single SDF file format by using open Babel to be imported in PyRx for virtual screening.

### Molecular docking

Flexible docking has been used in our studies to see the interactions of small molecules against DENV NS3 protease. The crystal structures were refined using PyMOL as crystal structures consist of other multiple chains, and errors like duplicate residues, and extra molecules like co-crystal ligands, water molecules etc. Water molecules and other ligand molecules were removed. The proteins were subjected to a minimization process. The minimization was carried out individually one by one in PDB_hydro sever, where the protein is minimized and the errors regarding side chain refinement, missing atoms adjustment is carried out. These minimized structures are used in the processes of molecular docking. For docking, AutoDockTools along with AutoDock Vina developed by the Scripps Research Institute has been used in our studies [29–31]. Total three molecules are selected as per the cut-off of the virtual screening. The AutoDock graphical interface AutoDockTools was used to add polar hydrogens and compute partial charges (Gasteiger charges) to the proteins and ligands. The search space was included in a box big enough to encompass the whole protein. Grid point spacing was 1.0 Å. An exhaustiveness of 50 was given to improve the quality of evaluation. The best poses have been identified on the basis of free energies and analysis of the interactions has been performed on the docked complexes.

## Virtual screening

Virtual screening has been performed in PyRx academic version, which was also developed by the Scripps Research Institute [32–36]. The prepared proteins (proteases) and ligands (library of small molecules) are imported into PyRx. The molecules imported are converted to Auto-Dock PDBQT files. AutoDock Vina as implemented in PyRx was used for the screening. Each protease is docked with the library of molecules at a time. Molecules are sorted according to their binding energies. A cut-off is set up for the molecules according to their free energies obtained from screening and leads were isolated to perform docking refinement studies as described above.

## Molecular dynamics (MD) simulations

MD was performed for type1 protease (PDB: 3L6P) bound with the three best hit molecules (dabrafenib, idelalisib and nintedanib) as shown in Scheme 2 and GROMACS (GROningen Machine for Chemical simulations) was used to perform the MD for duration of 100 ns using GROMOS96 53A6 force field. The lone protein was also subjected to MD simulation as control in order to gain information about the allosteric modulation on comparison with the results on the complexes. Standard protocol was followed for MD simulations [37].

**Scheme 2. Chemical structures of the three hit molecules as obtained by virtual screening against DENV protease**.

## Results and discussion

In order to find potential inhibitors that can bind and alter the proteins conformation we performed virtual screening and binding analysis of a library of drug molecules used as medicine for other human diseases as potential repurposed drug for DENV NS3 protease. 24 crystal structures of dengue NS3 protein from four different serotypes were retrieved from the PDB and are listed in S2 Table. Among them, 13 crystal structures contained protease domain (ten of them are composed of only the protease domain, and remaining three (2WZQ, 2VBC, 2WHX) structures are complete NS3 protein with helicase domain as well). The sequence analysis of protease domain of all the proteins (five serotypes) was carried out using Clustal Omega to analyze the conserved and active sites residues. A sequence homology of 67–71% was found between the DENV types 1–4 (Fig 1 and S1 Fig). The percent identity matrix in S1 Fig depicts that the serotype 2 shares an average of 67% homology with other serotypes. Type 4 and type 4T share nearly 99% of homology with each other and an average of 77% homology with the rest of the serotypes. Type 3 and type 1 share 85% of homology with each other and a homology of 78% with the rest of the serotypes.

To find leads for NS3 protease inhibitors, a total of 175 virtual screening calculations were performed. The lead molecules were identified purely on the basis of binding energy (free energy from docking) obtained from the results of virtual screening (S2 Fig). A cut-off is set up for the molecules to perform further studies. Molecules which showed binding energies less than -9 kcal/mol with almost all serotypes are selected for performing refinement docking to analyze the binding conformation of the ligand in the docked complexes. This lead to identify three drug molecules by screening method as discussed in the material and method sections. A detail docking analysis was carried out with crystal structures of the five proteases (i.e., one from each serotype with PDB IDs 3L6P (type 1), 4M9T (type 2), 3U1J (type 3), 2VBC (type 4), and 2WHX (type 4T)). The molecular crystals were chosen based on their resolution, model geometry and fit model data [24, 38]. Superimposition of the five crystal structures highlighting the active site is shown in (Fig 2). The key active site residues remained conserved throughout the evolution. The structures of the proteases were validated accordingly with the accepted regions as per Ramachandran plot (S3 Fig). The DENV protease also shows an allosteric ligand-binding site. In order to find the binding site, blind docking was performed followed by

**Fig 1. Multiple sequence alignment of DENV protease from five serotypes with highlighted active site residues.** Vide S1 Fig for percent identity matrix.

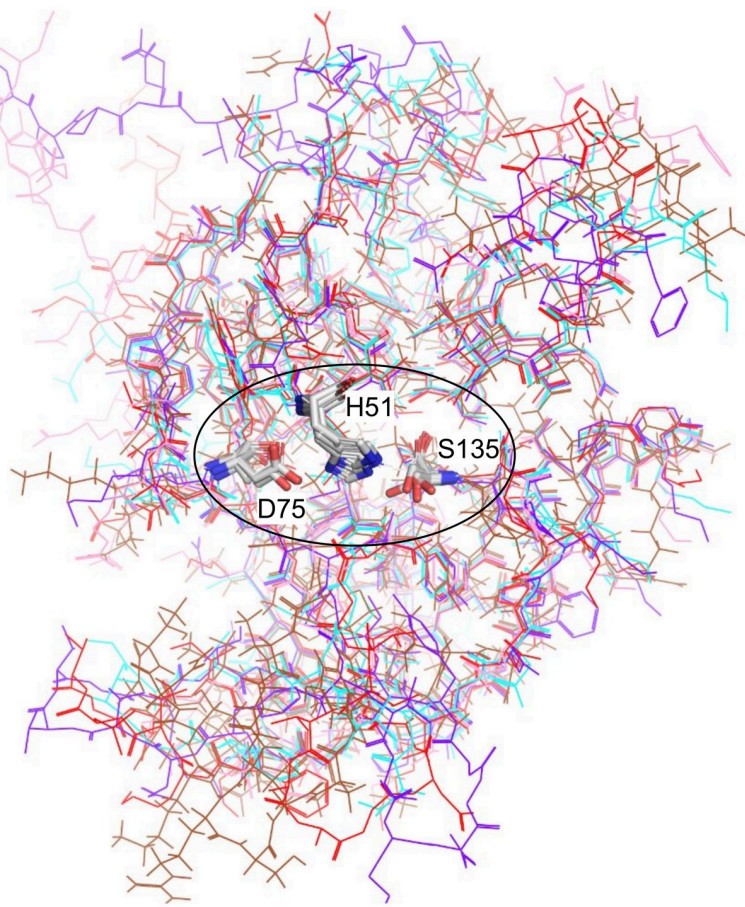

**Fig 2. Superimposed view of the protease structures from all five serotypes.** The three conserved active site residues, H51, D75 and S135 at the center are shown in stick model and encircled with a black outline. Color key for line models: cyan, type 1 (PDB: 3L6P); brown, type 2 (PDB: 4M9T); violet, type 3 (PDB: 3U1J); red, type 4 (PDB: 2VBC); pink, type 4T (PDB: 2WHX).

clustering of the docking results. From the docking we observed that the molecules that showed lowest binding energies did not bind to substrate-binding active site of the protease, instead the molecules favored non-active site on the enzyme. The site of the binding was allosteric site as reported in other studies [15, 16]. The comparison of the binding site across different serotype proteins was also identified and it was in the same region as shown in Fig 3. The best poses were identified on the basis of binding free energy (Table 1). The binding interactions and the affinity of the molecules to the proteases of all five serotypes of the DENV are discussed in the subsequent paragraphs.

Dabrafenib shows best binding energy for all the five serotypes with lowest binding energy of -9.1 kcal/mole (Fig 4a and Table 1). Dabrafenib, is sold under the brand name of Tafinlar. It is a drug for the treatment of cancers associated with a mutated version of the gene *BRAF* and it is often used as an inhibitor of the associated enzyme B-Raf, which plays a role in the regulation of cell growth. The molecule preferred the binding at the allosteric site of the DENV protease. Hydrophobic interaction was dominant across all the serotypes. In DENV2 protease, both hydrogen bonding and charged ionic interactions were also observed (Fig 4b). It was known that Lys73, Lys74, Trp83, Gln88, Thr120, Ile123, Gly124, Leu149, Asn152, Ala166, Asn167 in this protease play some key roles in the allosteric inhibition of the active site when

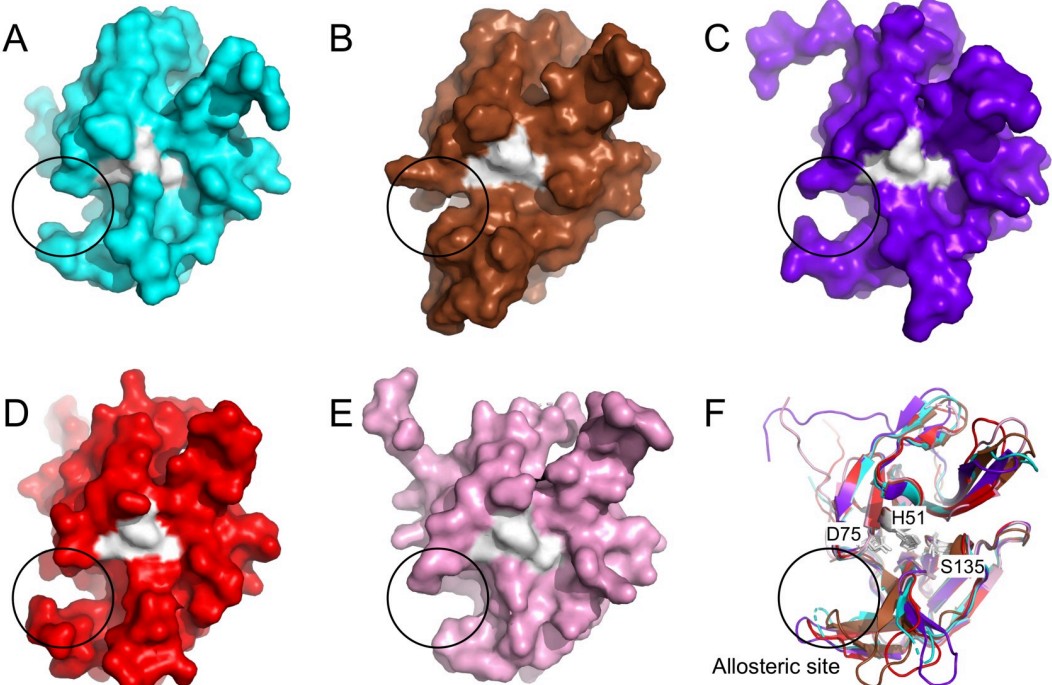

**Fig 3. Comparison of allosteric sites in DENV protease of different serotypes of dengue in surface representation.** (A-E) Type 1–4 and 4T of DENV protease. The allosteric binding pocket is shown with a circle. Color key: cyan (A), type 1 (PDB: 3L6P); brown (B), type 2 (PDB: 4M9T); violet (C), type 3 (PDB: 3U1J); red (D), type 4 (PDB: 2VBC); pink (E), type 4T (PDB: 2WHX);. (F) Superimposed view (ribbon) of all the five proteases from different serotypes.

binding with ligand by bringing out significant conformational changes and disabling the protein to make its function [39]. In our investigation, it was found that, residues Lys73, Lys74, Thr118, Glu120, Val123, Asn152 and Ala164 of the DENV1 protease significantly interact with all three molecules. These residues may perform a key role in allosteric modulation of DNV NS3 protease (Table 2). Further, we compared the interacting residues of allosteric sites of different dengue serotypes with dabrafenib molecule. We found that Lys73 and Lys74 were common among Type 1 and Type 2 DNV serotypes. Whereas in dengue type 4 and 4T, first interacting residue was Arg73 instead of Lys73. The presence of basic residue may be evolutionary insertion at position 73 of the allosteric sites of the NS3 proteases of all serotypes of Dengue for a proper interaction with binding molecule (Table 3). Hydrogen bonding interaction with amine functional group of the dabrafenib was also noticed with three proteases. This molecule shows pi stacking interactions with type 1 and 3 proteases.

**Table 1. Flexible docking binding energies of all serotypes of proteins with ligand molecules.**

| Proteins | Binding energies(ΔKcal/mol) | | |
|---|---|---|---|
| | Dabrafenib | Idelalisib | Nintedanib |
| Type 1(3L6P) | -8.6 | -8.6 | -8.6 |
| Type 2(4M9T) | -9.1 | -8.5 | -8.3 |
| Type 3(3U1J) | -9.2 | -8.4 | -8.2 |
| Type 4(2VBC) | -8.5 | -8.2 | -8.5 |
| Type 4 Thailand(2WHX) | -8.1 | -8.4 | -8.1 |

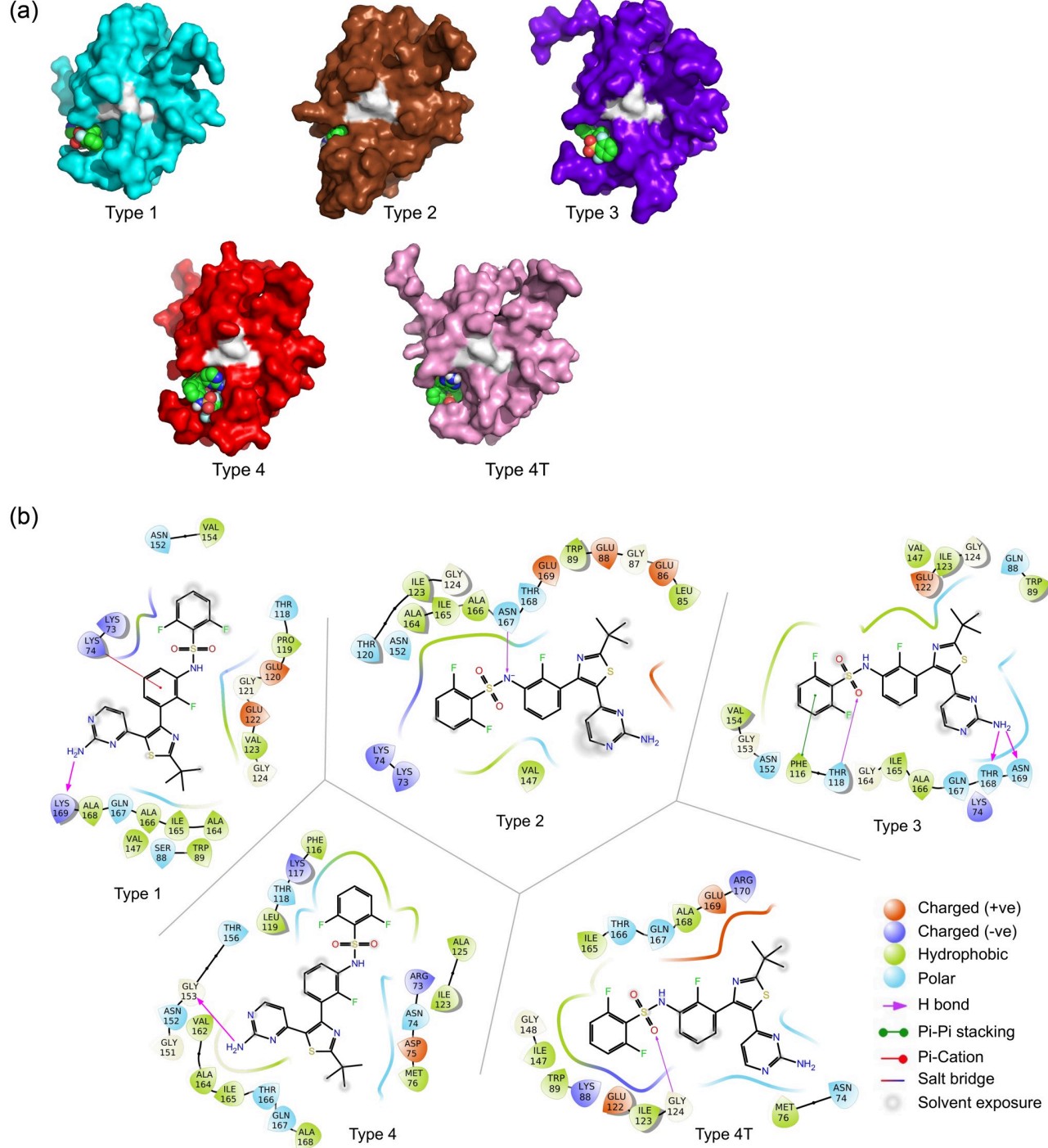

**Fig 4.** a. Graphical representation of detailed surface interaction of dabrafenib with the allosteric binding site residues of all five DENV proteases as obtained by molecular docking. b. 2D interaction diagram of dabrafenib with the binding site residues of all five DENV proteases as obtained by molecular docking. (A) Interaction of dabrafenib with the binding site residues of type 1 DENV protease (PDB: 3L6P). (B) Interaction of dabrafenib with the binding site residues of type 2 DENV protease (PDB: 4M9T). (C) Interaction of dabrafenib with the binding site residues of type 3 DENV protease (PDB: 3U1J). (D) Interaction of dabrafenib with the binding site residues of type 4 DENV protease (PDB: 2VBC). (E) Interaction of dabrafenib with the binding site residues of type 4T DENV protease (PDB: 2WHX).

**Table 2. Interacting allosteric site residues of DNV type I with all three inhibitory molecules.**

| Sr. No | Dabrafenib | Idelalisib | Nintedanib |
|---|---|---|---|
| 1 | Lys73 | Val72 | Lys73 |
| 2 | Lys74 | Lys73 | Lys74 |
| 3 | Ser88 | Lys74 | Ser88 |
| 4 | Trp89 | Asp75 | Phe116 |
| 5 | Thr118 | Phn116 | Thr118 |
| 6 | Pro119 | Thr118 | Pro119 |
| 7 | Glu120 | Glu120 | Glu120 |
| 8 | Val123 | Val123 | Glu122 |
| 9 | Gly124 | Asn152 | Val123 |
| 10 | Val147 | Gly153 | Ala125 |
| 11 | Asn152 | Val154 | Asn152 |
| 12 | Ala164 | Val162 | Val154 |
| 13 | Ile165 | Ser163 | Val162 |
| 14 | Ala166 | Ala164 | Ser163 |
| 15 | Gln167 | | Ala164 |
| 16 | Ala168 | | Ile165 |
| | | | Gln167 |
| | | | Ala168 |
| | | | Lys169 |
| | | | Ala170 |

Note- Similar residues are indicated with same colour.

Similar to dabrafenib, idelalisib which act as a phosphoinositide 3-kinase inhibitor, also showed good binding energies at allosteric sites (Table 1). Idelalisib is sold under the brand name Zydelig. It is a medication used to treat certain blood cancers. The binding of idelalisib was also dominated by hydrophobic interaction, however no hydrogen bonding interaction or pi stacking was noticed (S4 Fig). Two ionic interactions were noticed, one with type 3 and another with type 4T variant proteases. In DENV4T, the molecule was showing charged ionic interactions with residues Arg73 and Lys88 with aromatic moiety of the ligand molecule.

The other potential drug candidate, nintedanib, also favored the binding to the allosteric site of the proteases with good binding affinities against all the serotypes (Table 1). The medicine is sold under the brand names Ofev and Vargatef, and used for the treatment of idiopathic pulmonary fibrosis. Hydrogen bonding was not noticed in nintedanib binding with type 2 protease. Pi-pi interaction was present with DENV3 and DENV4 proteases. However, pi-cation interaction was seen in DENV3, DENV4, and DENV4T variants. The detailed interaction diagram of nintedanib is shown in S5 Fig.

In order to gain further insight into the mechanism of how ligand binding into the allosteric site of DENV protease impairs its function, MD simulations were performed on the three-ligand protease complexes (three ligand and DENV1 protease with PDB ID 3L6P). Dynamics of free protein was carried out as a control. Fig 5 shows the overall changes (standard deviations) in the protein (3L6P) structure upon binding to the molecules to the free protein over the course of simulation. It was observed that the protein was stable in a dynamic state and system was equilibrated and the deviations range fell below t3 Å, suggesting that large structural perturbation did not occur in free protein or after the allosteric ligand binding.

Radius of gyration ($R_g$) is the distribution of atoms of protein around its center of mass. It is used to analyses the overall compactness of a macromolecule. $R_g$ of DENV1 protease

**Table 3. Interaction of dabrafenib with all 5 DNV serotypes.**

| Sr. No | DENV 1 | DENV2 | DENV3 | DENV4 | DENV4T |
|---|---|---|---|---|---|
| 1 | Lys73 | Lys73 | Lys74 | Arg73 | Arg73 |
| 2 | Lys74 | Lys 74 | Gln88 | Asn74 | Asn74 |
| 3 | Ser88 | Leu85 | Trp89 | Asp75 | 76Met |
| 4 | Trp89 | Glu86 | Phe116 | Met76 | Lys88 |
| 5 | Thr118 | Gly87 | Thr118 | Phe116 | Trp89 |
| 6 | Pro119 | Glu88 | Glu122 | Lys117 | Lys 91 |
| 7 | Glu120 | Trp89 | Ile123 | Thr118 | Leu115 |
| 8 | Val123 | Thr120 | Gly124 | Leu119 | Glu122 |
| 9 | Gly124 | Ile123 | Val147 | Ile123 | Ile123 |
| 10 | Val147 | Ala125 | Asn152 | Ala125 | Gly124 |
| 11 | Asn152 | Val147 | Ile165 | Gly151 | Ile145 |
| 12 | Ala164 | Asn152 | Ala166 | Asn152 | Thr166 |
| 13 | Ile165 | Val162 | Gln167 | Gly153 | Gln167 |
| 14 | Ala166 | ALa164 | Thr168 | Thr156 | Ala168 |
| 15 | Gln167 | Ala166 | Asn169 | Val162 | Glu169 |
| 16 | Ala168 | Asn167 |  | Ala164 |  |
| 17 |  | Thr168 |  | Ile165 |  |
| 18 |  | Glu169 |  | Thr166 |  |
| 19 |  |  |  | Gln167 |  |

complexes were compared with that of the free protein and it is shown in S6 Fig. $R_g$ values significantly decreased upon binding to dabrafenib. If a protein unfolds, the $R_g$ of the protein increases. Decrease in $R_g$, in turn, indicated structural stability. However, it remained the same for nintedanib binding resonating the less effectiveness of nintedanib as an allosteric ligand; whereas dabrafenib appeared to be more effective in inducing structural changes in the target protease.

Local fluctuations in the protein are measured by root mean square fluctuation (RMSF) and they are useful for characterizing local changes along the protein chain [40]. RMSF of DENV1 protease upon binding to the three ligands was compared with that of the free protein in Fig 6. The peaks in the plot indicate amino acid residues of the protein that fluctuate the most during the MD simulation. Typically, the tails (N- and C-terminals) fluctuate more than any other part of the protein. Secondary structure elements like alpha helices and beta strands are usually more rigid than the unstructured part of the protein, and thus fluctuate less than the loop regions. The regions with Leu30, Glu61, Leu119, and Ser158 showed fluctuations below 3 Å range. These are present in the important loop regions of the protein. In protein-dabrafenib complex (Fig 6) the rate of fluctuations are higher at residue no 39 when compared to protein alone; whereas at no 50 residue the rate of fluctuations are higher in protein compared to that of the complex. In other residues, the fluctuations between protein and complex remained almost the same. In protein-idelalisib complex, the rates of fluctuations are higher at 55 residue than that of the protein. In all remaining residues the rate of fluctuations in complex were less compared to that of in the free protein. In protein-nintedanib complex, the rate of fluctuations was higher at no 41 residue than the free protein. In all remaining residues the rate of fluctuations in complex were less compared to the protein alone.

Hydrogen bond interactions also play crucial role in biomolecular recognition [41]. Hydrogen bond profiles between the selected molecules and the target protein DENV1 (PDB: 3L6P)

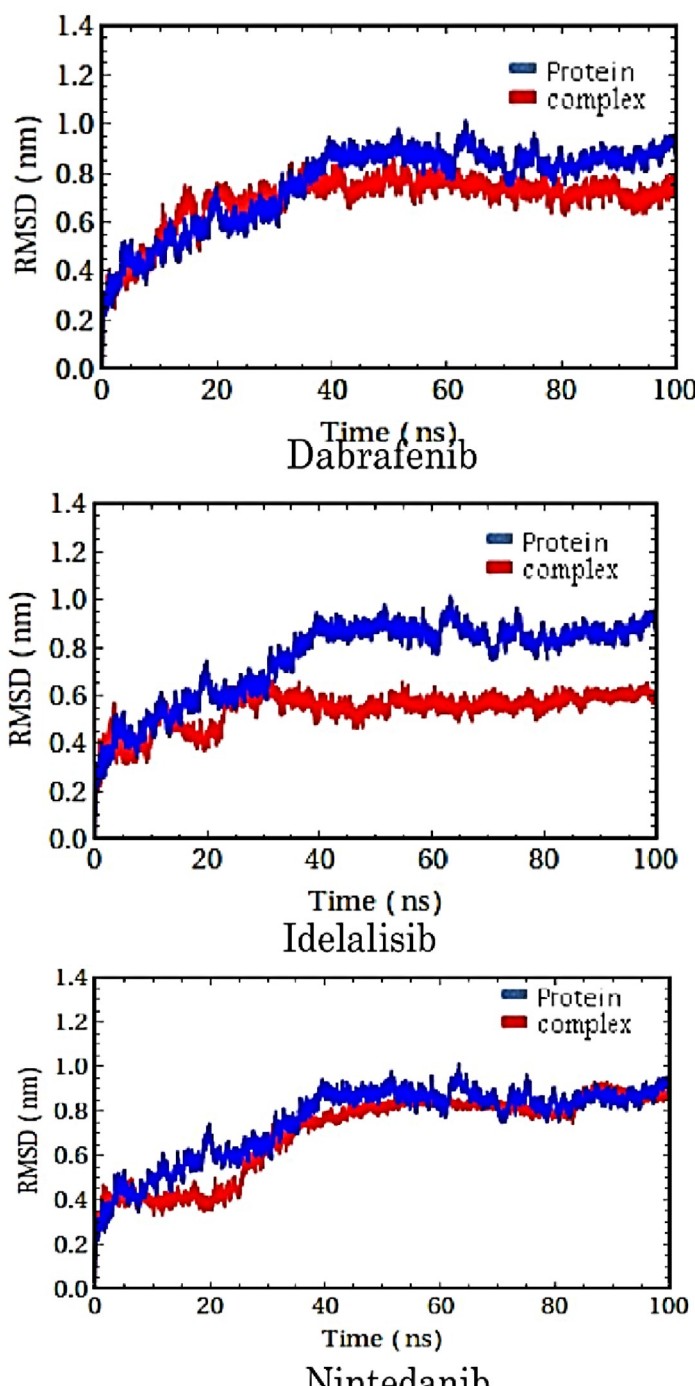

**Fig 5. Overall structural changes in the type 1 dengue protease upon binding to the allosteric ligands.**

were calculated from the MD simulation trajectory. It was found that dabrafenib (Fig 7) did not form hydrogen bonds with the protein initially. However, after the molecular recognition occurred, it formed 3 hydrogen bonds indicating induced fit mode of binding. It is observed that idelalisib (Fig 7) can form one hydrogen bond by the end of simulation whereas 2 hydrogen bonds are formed in the early period of simulation. Nintedanib was found to form as

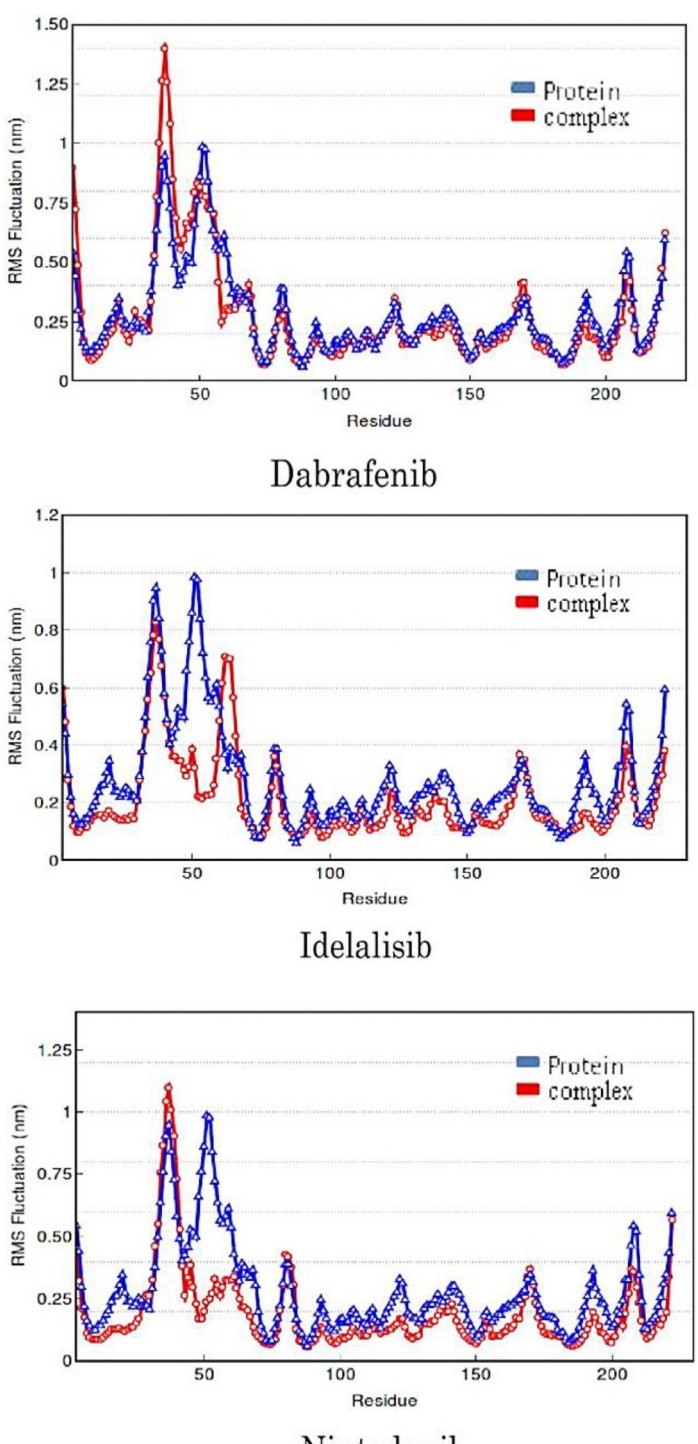

**Fig 6. Residue-wise fluctuations in the type I dengue protease upon binding to the allosteric ligands.**

many as 5 hydrogen bonds by the end of simulation whereas it showed only 1 hydrogen bond in the docked structure (S5 Fig).

Further, the effect of allosteric ligand binding on the secondary structure of DENV1 NS3 protease was analyzed using DSSP (Dictionary of Secondary Structure of Proteins) tools. It

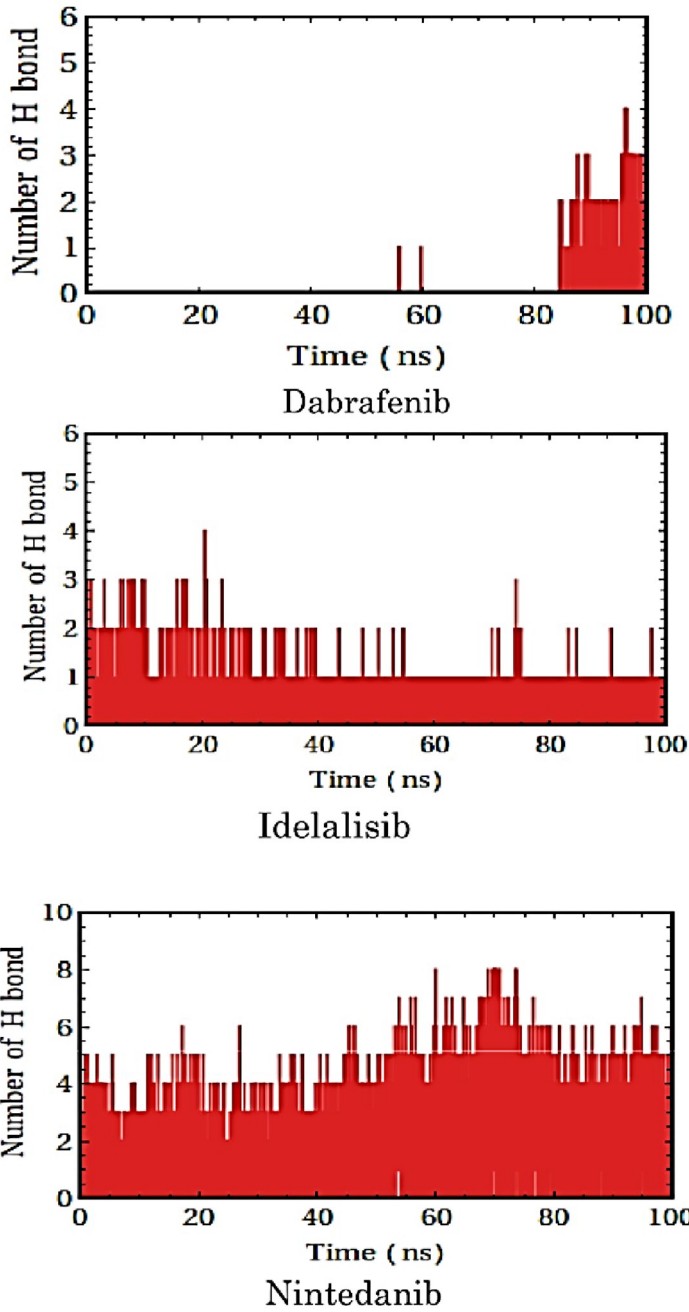

**Fig 7. Hydrogen bonding in the molecular recognition of the allosteric ligands with the allosteric binding pocket of the type 1 dengue protease.**

provides rough estimation of protein secondary structures along the time in MD simulations trajectory. Profile of secondary structure of protein and their complexes along the trajectory are shown in S7 and S8 Figs. In the dabrafenib-protein complex, there was an increase in bends from 20–30 residues region of the protein. An α -helix were found in the complex after 60 ns in between residues 55–65 whereas α-helix is seen in protein at 30ns at same residues. Bends in protein at residues 60–80 were replaced by coils in complex. The number of β-sheets increased in the complex between 160–180 residues after 70 ns of simulation. Fluctuations in

bends and turns are more in the complex after 80 ns beyond 200th residues. Stable β-sheet was seen in the complex at around 200th residue whereas it is not observed in free protein. In protein-idelalisib complex, it was observed that α-helix is formed at 55–60 residues. Number of bends between 60–80 residues decreased in the complex. α-helix around residue 110, which was stable in protein got disrupted in the complex. The bends which were fluctuating in protein between 150–155 residues became stable in the complex. β-sheet region after residue 200 in complex was increased compared to the protein. An α-helix at 55–60 residues in protein was not seen in the protein-nintedanib complex. After 80th residue, turns and bends that were fluctuating got converted to bends in the complex.

Out of the three molecules, nintedanib exerted the least changes in RMSD, $R_g$ and RMSF and the secondary structure compared to the free protein. Low RMSD value indicates that the structure of protein was least perturbed by ligand binding and the complex was very stable. Therefore, nintedanib provide good stability to the DENV1 NS3 protease. On the other hand, most structural perturbation in the protein was observed upon its binding to idelalisib, thus, indicating most potent effect idelalisib as an allosteric ligand. Dabrafenib also showed potent allosteric effect on the DENV1 NS3 protease in terms of structural perturbations.

NS3 protease is a trypsin-like serine protease, performs the post-translational proteolysis processing of the precursor protein. Therefore, several attempts have been made to find inhibitors against DENV by targeting the enzyme using both in *in-silico* and *in vivo* methods. The importance of proteins motion and fluctuation in the allosteric sites have significant impacts on the functional efficacy of proteases [23, 24]. Upon small-molecule binding to the allosteric sites, the protein undergoes a change in conformation or an alteration in the conformational equilibrium that affects the function of the enzyme. Mukhametova et al. have explored the allosteric binding site through large-scale virtual screening and molecular dynamics simulations followed by calculations of binding free energy [39]. They have proposed two mechanisms for enzyme inhibition: firstly, ligand may either destabilize electronic density or create steric effects relating to the catalytic triad residues His51, Asp75, and Ser135; secondly, the ligand may disrupt the movement of the C-terminal of NS2B required for inter-conversion between the "open" and "closed" conformations of the enzyme.

We observed distinct conformational changes in the protein structure upon the ligand binding except for nintedanib. It indicated an induced fit mode of binding for dabrafenib and idelalisib, whereas the binding of nintedanib was mostly conformational selection. MD simulation, thus, indicated that nintedanib may not be effective modulator for the enzyme, as it cannot impart structural changes in the target protein. However, MD reaffirms the potential of the other two molecules to be allosteric modulators of DENV1 protease as they can impart significant structural changes upon binding to the target protein.

## Conclusion and perspectives

Dengue virus is the cause of dengue fever and dengue shock syndrome (DSS). However, no specific antiviral treatment for the dengue infection is available so far. In recent years, a considerable research effort is being directed to discover therapies and drugs for the disease and several molecules are suggested to target the active site of NS3 protease. It is a key enzyme involves in viral replication and human infection. Therefore, the inhibition of this protease activity is an important route for preventing viral infection. In our current investigation, we could establish three known drug molecules, dabrafenib, idelalisib and nintedanib could effectively bind to the allosteric site and affect the dynamics and conformation of the enzyme. This may cause modulaton of the functional activity. The potential allosteric site on NS3 protease was characterized and the active residues were identified.

The allosteric sites are evolutionarily less conserved and tend to vary in the different sero-types thus giving the opportunity to develop target specific drugs. Identification of the allosteric site also offers non-peptide based drug development opportunities for the protease, for which substrate analogue peptide based drugs are generally designed that can compete with the peptide substrate and strongly bind into the active site. However, peptide drugs are more susceptible to proteolytic digestion, thus, non-peptide based allosteric modulator provides and alternative strategy for the drug development for DENV proteases. Thus, the binding of the molecules could alternately terminate or decrease the performance of protease function. Experimental studies, however, may justify if the allosteric modulation favor/disfavor the activity of the proteases upon the binding of the molecules and whether it is to be considered as a drug candidate for dengue fever and related diseases.

## Supporting information

**S1 Fig. Percentage identity matrix for the multiple sequence alignment of the proteases of five serotypes as generated by ClustalOmega.**
(TIF)

**S2 Fig. Matrix plot showing the binding energies of the drug molecules against the five different serotypes as obtained as leads by virtual screening.**
(TIF)

**S3 Fig. Ramachandran plots of selected proteases for docking and molecular dynamic analysis.**
(TIF)

**S4 Fig. Detailed interaction diagram of idelalisib with the binding site residues of all five proteases as obtained by molecular docking.**
(TIFF)

**S5 Fig. Detailed interaction diagram of nintedanib with the binding site residues of all five proteases as obtained by molecular docking.**
(TIFF)

**S6 Fig. Dynamic changes in the gyrating radius of the type 1 protease upon binding to the different synthetic allosteric ligands.**
(TIF)

**S7 Fig. Secondary structure of free type 1 dengue protease over the course of MD simulation.** Color key: white, coil; red, beta-strand; black, beta-bridge; green, bend; yellow, turn; blue, alpha-helix; purple, pi-helix; gray, 3–10 helix.
(TIF)

**S8 Fig. Changes in the secondary structure of type 1 dengue protease upon binding to the allosteric ligands: Dabrafenib, idelalisib, nintedanib.** Color key: white, coil; red, beta-strand; black, beta-bridge; green, bend; yellow, turn; blue, alpha-helix; purple, pi-helix; gray, 3–10 helix.
(TIF)

**S1 Table. List of the drug molecules along with their accession codes.**
(DOCX)

**S2 Table. Available crystal structures of DENV NS3 protein.** PDB IDs marked with * are the 3D structures containing protease domain. Note:—1,2,3,4, refers to DENV types and 4T refers

to DENV 4 Thailand variety.
(DOCX)

## Author Contributions

**Conceptualization:** Achintya Saha, Uttam Pal, V. Ravichandiran, Nakul C. Maiti.

**Data curation:** R. V. Sriram Uday.

**Formal analysis:** Rajdip Misra.

**Investigation:** Rajdip Misra, Nakul C. Maiti.

**Methodology:** R. V. Sriram Uday, Annaram Harika.

**Software:** R. V. Sriram Uday, Annaram Harika, Sandip Dolui.

**Supervision:** Nakul C. Maiti.

**Validation:** R. V. Sriram Uday, Sandip Dolui, Achintya Saha, Uttam Pal, V. Ravichandiran.

**Writing – original draft:** Rajdip Misra, Nakul C. Maiti.

**Writing – review & editing:** Rajdip Misra, Uttam Pal, Nakul C. Maiti.

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
