## [Decision Letter · Decision Letter 0]

30 Jun 2021

PONE-D-21-11598

Dabrafenib Acts as a Significant Allosteric Modulator for Dengue NS3 Protease

PLOS ONE

Dear Dr. Maiti,

Thank you for submitting your manuscript to PLOS ONE. After careful consideration, we feel that it has merit but does not fully meet PLOS ONE’s publication criteria as it currently stands. Therefore, we invite you to submit a revised version of the manuscript that addresses the points raised during the review process.

We look forward to receiving your revised manuscript.

Kind regards,

Keivan Zandi

Academic Editor

PLOS ONE

Journal Requirements:

4. We note that Figures 3, 4a and Scheme 1 in your submission contain copyrighted images. All PLOS content is published under the Creative Commons Attribution License (CC BY 4.0), which means that the manuscript, images, and Supporting Information files will be freely available online, and any third party is permitted to access, download, copy, distribute, and use these materials in any way, even commercially, with proper attribution. For more information, see our copyright guidelines: http://journals.plos.org/plosone/s/licenses-and-copyright.

a. You may seek permission from the original copyright holder of Figures 3, 4a and Scheme 1 to publish the content specifically under the CC BY 4.0 license. 

Additional Editor Comments:

Reviewers' comments:

Reviewer's Responses to Questions

**Comments to the Author**

1. Is the manuscript technically sound, and do the data support the conclusions?

Reviewer #1: Yes

Reviewer #2: Yes

2. Has the statistical analysis been performed appropriately and rigorously? 

Reviewer #1: Yes

Reviewer #2: Yes

3. Have the authors made all data underlying the findings in their manuscript fully available?

Reviewer #1: Yes

Reviewer #2: Yes

4. Is the manuscript presented in an intelligible fashion and written in standard English?

Reviewer #1: Yes

Reviewer #2: Yes

5. Review Comments to the Author

Reviewer #1: The authors present a thorough in situ modeling approach to predict potential of small peptides to competitively bind to Dengue NS3 non-active site (allosterically) using routine molecular modeling tools. The authors identify 3 lead small molecules based upon their delta KCal/mol values of binding (< -9 kcal/mol): Dabrafenib, idelalisib and nintedanib.

As there is no current approved therapy or antiviral (small molecule) for Dengue, a positive control was not available. However, authors referenced in conclusions previous studies with prochlorperazine (PCZ) which also had in vitro data supporting possible antiviral activity for Dengue (targeting NS3 protease). Why was PCZ not included in initial screening as control to validate modeling assays to others in the field? Notably, in Table 2, suggest include PCZ for comparison to 3 "hit" compounds.

Secondly, why do authors focus on Dabrafenib (title) when appears 3 hits were identified. Also, could be argued that based upon FIg 5, Idelalisib is predicted to have most impact on altering structure of NS3 protease (via allosteric binding). Caution that focus on Dabrafenib suggests conflict of interest or personal interests.

Third, suggest grammar/text be reviewed by expert editor as several spacing, capitalization, spacing and subscript use is inconsistent. Most notably, Fig 6 has mislabel of top figure (labeled twice as idelalisib, suggest it should be "dabrafenib".).

Lastly, as this is only predictive, clearly in vitro work is necessary follow up and more relevant going forward to confirm antiviral potential and more importantly to determine potential toxicity. As these are anticancer drugs, toxicity is often associated. targeting protease is likely to have non-specific interaction with host proteases as well. Must test for potential tox.

Reviewer #2: the authors try to established three drug molecules (dabrafenib, idelalisib and nintedanib)against all the dengue serotypes (dengue1, dengue 2, dengue 3, dengue 4 & dengue 4T) protease. All the three drugs candidates show stable and strong binding with dengue NS3 based on the authors analysis. The author has done sufficient in silico study to investigate potentially of three drugs inhibiting dengue virus through dengue protease (NS3/NS2B). I have no objection to proceed this manuscript for publishing.

6. PLOS authors have the option to publish the peer review history of their article (what does this mean?). If published, this will include your full peer review and any attached files.

Reviewer #1: No

Reviewer #2: **Yes: **Pouya Hassandarvish

---

## [Author Response · Author response to Decision Letter 0]

10 Aug 2021

Response to the questions and queries raised by PLOS ONE editor and reviewers: 

Author response: All relevant data are given within the manuscript and the supporting documents. We don’t have any other data to provide as repository information. 

Author response: We have added the ORCID iD of corresponding author. 

4. We note that Figures 3, 4a and Scheme 1 in your submission contain copyrighted images. All PLOS content is published under the Creative Commons Attribution License (CC BY 4.0), which means that the manuscript, images, and Supporting Information files will be freely available online, and any third party is permitted to access, download, copy, distribute, and use these materials in any way, even commercially, with proper attribution. For more information, see our copyright guidelines: http://journals.plos.org/plosone/s/licenses-and-copyright.

Author response: We have regenerated Figures 3, 4a and Scheme 1 and replaced them accordingly. We have also modified Figures 1, 2, 4b and scheme 2 with better graphics. 

Author response: Captions for Supporting Informations have been added accordingly. 

Additional Editor Comments:

Please review your reference list to ensure that it is complete and correct. If you have cited papers that have been retracted, please include the rationale for doing so in the manuscript text, or remove these references and replace them with relevant current references. Any changes to the reference list should be mentioned in the rebuttal letter that accompanies your revised manuscript. If you need to cite a retracted article, indicate the article’s retracted status in the References list and also include a citation and full reference for the retraction notice

Author response: Reference list has been reviewed thoroughly. We have modified the list and cited some additional research articles for better understanding. 

The following references have been added: 

1. Kharisma V, Muhammad A, Probojati R, Tamam M, Antonius Y. Revealing Potency of Bioactive Compounds as Inhibitor of Dengue Virus (DENV) NS2B/NS3 Protease from Sweet Potato (Ipomoea batatas L.) Leaves. Indian J Forensic Med Toxicol. 2021;15: 1627. doi:10.37506/ijfmt.v15i1.13644

2. Dwivedi VD, Bharadwaj S, Afroz S, Khan N, Ansari MA, Yadava U, et al. Anti-dengue infectivity evaluation of bioflavonoid from Azadirachta indica by dengue virus serine protease inhibition. J Biomol Struct Dyn. 2021;39: 1417–1430. doi:10.1080/07391102.2020.1734485

3. Panda K, Alagarasu K, Patil P, Agrawal M, More A, Kumar NV, et al. In Vitro Antiviral Activity of α-Mangostin against Dengue Virus Serotype-2 (DENV-2). Molecules. 2021;26: 3016. doi:10.3390/molecules26103016

4. Millies B, von Hammerstein F, Gellert A, Hammerschmidt S, Barthels F, Göppel U, et al. Proline-Based Allosteric Inhibitors of Zika and Dengue Virus NS2B/NS3 Proteases. J Med Chem. 2019;62: 11359–11382. doi:10.1021/acs.jmedchem.9b01697

5. Hassandarvish P, Rothan HA, Rezaei S, Yusof R, Abubakar S, Zandi K. In silico study on baicalein and baicalin as inhibitors of dengue virus replication. RSC Adv. 2016;6: 31235–31247. doi:10.1039/C6RA00817H

6. Protein-Ligand Complex. [cited 5 Aug 2021]. Available: http://www.mdtutorials.com/gmx/complex/index.html

5. Review Comments to the Author

Reviewer #1: The authors present a thorough in situ modeling approach to predict potential of small peptides to competitively bind to Dengue NS3 non-active site (allosterically) using routine molecular modeling tools. The authors identify 3 lead small molecules based upon their delta KCal/mol values of binding (<-9 kcal/mol): Dabrafenib, idelalisib and nintedanib.

Comment: As there is no current approved therapy or antiviral (small molecule) for Dengue, a positive control was not available. However, authors referenced in conclusions previous studies with prochlorperazine (PCZ) which also had in vitro data supporting possible antiviral activity for Dengue (targeting NS3 protease). Why was PCZ not included in initial screening as control to validate modeling assays to others in the field? Notably, in Table 2, suggest include PCZ for comparison to 3 "hit" compounds.

Author response: We thank the reviewer for raising this issue. Prochlorperazine (PCZ) has been reported as an entry blocker/ inhibitor and not a NS3 protease inhibitor of DENV. As the proposed mechanism of action and the target protein of PCZ is completely different, we did not include PCZ as control. We have clarified this issue in the revised manuscript.

Comment: Secondly, why do authors focus on Dabrafenib (title) when appears 3 hits were identified. Also, could be argued that based upon FIg 5, Idelalisib is predicted to have most impact on altering structure of NS3 protease (via allosteric binding). Caution that focus on Dabrafenib suggests conflict of interest or personal interests.

Author response: It is true that according to Fig 5. Idelalisib seems to be more impactful on alteration of DENV NS3 protein structure. On contrary, dabrafenib showed better (lowest) binding energies (kcal/mol) against all of the dengue serotypes. From S6 Fig, it can also be seen that upon binding to dabrafenib, radius of gyration value (Rg) significantly decreased. However we want to declare that there is no conflict of interest or personal interest from our end. We have changed the title of the revised manuscript to accommodate the three identified hits. 

Comment: Third, suggest grammar/text be reviewed by expert editor as several spacing, capitalization, spacing and subscript use is inconsistent. Most notably, Fig 6 has mislabel of top figure (labeled twice as idelalisib, suggest it should be "dabrafenib".).

Author response: We thank the reviewer for pointing out the mistake. We have corrected the labels of Fig 6 in the revised manuscript. We have also thoroughly revised the text to rectify any typos, inconsistent formats and grammatical mistakes.

Comment: Lastly, as this is only predictive, clearly in vitro work is necessary follow up and more relevant going forward to confirm antiviral potential and more importantly to determine potential toxicity. As these are anticancer drugs, toxicity is often associated. targeting protease is likely to have non-specific interaction with host proteases as well. Must test for potential tox.

Author response: We thank the reviewer for the suggestion. In future we plan perform in-vitro assays and cytotoxicity tests against all three potential drug candidate in collaboration with experimental research groups.

---

## [Editor Report · Decision Letter 1]

26 Aug 2021

Dabrafenib, Idelalisib and Nintedanib Act as Significant Allosteric Modulator for Dengue NS3 Protease

PONE-D-21-11598R1

Dear Dr. Maiti,

We’re pleased to inform you that your manuscript has been judged scientifically suitable for publication and will be formally accepted for publication once it meets all outstanding technical requirements.

Kind regards,

Keivan Zandi

Academic Editor

PLOS ONE
---

## [Editor Report · Acceptance letter]

31 Aug 2021

PONE-D-21-11598R1 

Dabrafenib, Idelalisib and Nintedanib Act as Significant Allosteric Modulator for Dengue NS3 Protease 

Dear Dr. Maiti:

I'm pleased to inform you that your manuscript has been deemed suitable for publication in PLOS ONE. Congratulations! Your manuscript is now with our production department. 

Kind regards, 

on behalf of

Dr. Keivan Zandi 

Academic Editor

PLOS ONE